

# **Diagnosis of dust- and haze pollution-impacted PM$_{10}$, PM$_{2.5}$,**
# **and PM$_{1}$ aerosols observed at Gosan Climate Observatory**

**Xiaona Shang[1], Meehye Lee[1*], Saehee Lim[1], Örjan Gustafsson[2]**

**Gangwoong Lee[3], Limseok Chang[4]**

[1] Department of Earth & Environmental Sciences, Korea University, Seoul, South Korea

[2] Department of Applied Environmental Science (ITM) and the Bolin Centre for Climate Research, Stockholm University, 10691 Stockholm, Sweden

[3] Department of Environmental Science, Hankuk University of Foreign Studies, Seoul, South Korea

[4] National Institute of Environmental Research (NIER), Incheon, South Korea

21  [*] Correspondence to: M. Lee (meehye@korea.ac.kr)



## Abstract

In East Asia, soil dust is a major component of aerosols and is mixed with various pollutants during transport, resulting in large uncertainty in climate and environmental impact assessment and relevant policymaking. To diagnose the influence of soil dust and anthropogenic pollution on bulk aerosol, we conducted long-term measurements of mass, water-soluble ions, and carbonaceous compounds of $PM_{10}$, $PM_{2.5}$, and $PM_1$ at Gosan Climate Observatory, South Korea, from August 2007 to February 2012. The principle component analyses of all measured species reveal that the impact of anthropogenic pollution, soil dust, and agricultural fertilizer accounts for 46%, 16%, and 9% of the total variance, respectively. Particularly, the loadings of agricultural component were high in the warmer months with the least occurrence of high concentration events and have increased over time. In mode analysis of $PM_{10}$, $PM_{2.5}$, and $PM_1$ mass concentrations, the mean + σ was comparable to the $90^{th}$ percentile and thus, suggested as a robust criterion that determines the substantial impact of soil dust and haze pollution on particulate matter. The results of this study imply that non-combustion sources such as soil dust will impose constraints to the reduction of $PM_{2.5}$ as well as $PM_{10}$ concentrations. In addition, questions are raised as to whether the yearly average concentration is suitable for environmental standard in northeast Asian region.



## Introduction

Dust particles are dominant atmospheric aerosols and account for more than 60% of the total global dry aerosol mass burden (Textor et al., 2006). The emission of dust ranges from 1000 to 3000 Tg yr$^{-1}$ (Zender et al., 2004). Dust particles are abundant in coarse mode, which is represented by $PM_{10}$. Recently, a new type of dust particle has been observed in submicron size that was long-range transported from dry lake deposits in northern China. These particles are more abundant in salts and mineral constituents compared with typical soil dust (Shang et al., 2018). The high-mass loading of dust adversely affects air quality and causes climate change. In regional to global earth's environment, mineral dust contributes to a radiative forcing of $-0.3 \sim +0.1$ W m$^{-2}$ with a large uncertainty (IPCC, 2007, 2013) because emission source, chemical and mineralogical composition, and particle size vary in a wide range (Choobari et al., 2014). Dust particles have also been predicted to reduce the rate of ozone production (Dickerson et al., 1997) and promote new particle formation and growth by mixing with other pollutants (Nie et al., 2014; He et al., 2014). Their role in climate change is further highlighted by modifying cloud microphysical process via cloud droplet activation (Bègue et al., 2015) and $CO_2$ uptake via ocean fertilization (Pabortsava, et al., 2017).

A large amount of dust is found over arid regions in North Africa, North China and South Mongolia (Choobari et al., 2014). In East Asian desert areas, soil type significantly varies from arid and semi-arid deserts to loess deposits and dry lake deposits, depending on the formation process and climate (Zhang et al., 2009). Dust particles generated from these regions are different in chemical and mineralogical composition (Cheng et al., 2012; Kunwar and Kawamura, 2014; Liu et al., 2011; Tao et al., 2014; Su and Toon, 2011), leading to different environmental effects. For example, African dust contains more iron oxides mineral in the form of hematite as compared with those from Asia (Formenti et al., 2011; 2014), implying greater light absorption (Zhang et al., 2015). Among atmospheric aerosols generated from variety of emissions (Geng et al., 2014). EC is a main species absorbing light and measured as black carbon (BC) (Han et al., 2010; Saleh et al., 2014). In Northeast Asia, aerosols from various sources are often mixed together while being transported (Kim et al.,



2007) and dust plumes have been identified as enhanced concentrations of sulfate, nitrate,
ammonium, OC or EC over the Korean peninsula (Shin et al., 2015).

An array of AERONET and satellite observations is a vital tool for investigating atmospheric
aerosol at regional and global scales (Zhuang et al., 1992; Zhang et al., 2003; Jin et al., 2016;
Nazari et al., 2016). Because these techniques measure the extinction of bulk aerosols, it is
crucial to accurately estimate the optical property of major aerosol constituents (e.g., Kim et
al., 2007) in order to assess the anthropogenic contribution to total aerosol burden and its
climate effect using remote sensing data. However, it is still a big challenge to estimate the
optical property of main aerosol types especially dust particles in East Asia, because their
property is not only dependent on source regions (Huang et al., 2014) but also modified
during transport through aging and mixing with pollutants (McFarlane et al., 1992; Von
Salzen et al., 2005; Bäumer et al., 2007; Kim et al., 2011). For instance, the mixture of dust
and pollutant caused a significant increase of radiative forcing by 0.06 $Wm^{-2}$ (Zhang et al.,
2013a; Mishra et al., 2010; Li et al., 2012). In this context, it is important to figure out the
extent of dust particles in the atmosphere and distinguish them from bulk aerosol particles.
For instance, the ratio of $PM_{10}/PM_{2.5}$ has been employed to eliminate the effect of soil dust
from $PM_{2.5}$ when determining the mass absorption coefficient of OC (Chung et al., 2012).

In this study, to diagnose the impact of airborne dust and anthropogenic pollution on
atmospheric particulate matter and find out feasible criteria, the five-year measurements of
mass, water-soluble ions, and carbonaceous compounds for $PM_{10}$, $PM_{2.5}$, and $PM_1$ were
analyzed using statistical methods. The long-term and composite measurements of particulate
matter in different sizes are scarce in the study region. The results of mass mode analysis and
principle component analysis will provide insight into the role and significance of mineral
dust and haze pollution on particulate matter and further its control strategies in northeast
Asia.






## 1 Methodology

Aerosol samples were collected separately for $PM_1$, $PM_{2.5}$, and $PM_{10}$ onto 37 mm Teflon and
Quartz filters (Pall, Corp.) using sharp-cut cyclones (URG, USA) at the Gosan Climate
Observatory (GCO) from 2007 to 2012. Sampling was undertaken for a period of 24 h from
10:00 to 10:00 the next day. A total of 152 sets of samples were collected and analyzed for
water-soluble inorganic ions and carbonaceous compounds. Details about the measurement
methodology can be found in Lim et al. (2012, 2014). During the five-year period, five dust
and eleven haze events were recorded across Korea by the Korea Meteorological
Administration (KMA) mostly during the cold seasons from late fall to spring (Fig. 1).
Teflon filters were conditioned for 24 h in desiccators (SANPLATEC, Japan) under a relative
humidity of approximately 30%–40% and weighed before and after sampling using an
analytical balance (Denver, Germany). Water-soluble species were extracted from the filters
into a solution comprising a mixture of 19 mL distilled water and 1 mL methanol. Water-
soluble ions, including $Cl^-$, $NO_3^-$, $SO_4^{2-}$, $Na^+$, $NH_4^+$, $K^+$, $Mg^{2+}$, and $Ca^{2+}$ were analyzed via
ion chromatography (IC 25, Dionex, USA). For this analysis, 500 μL of sample was injected
by an auto-sampler into the AG11 and AS11 columns for anions or CG11 and CS11 columns
for cations (Dionex, USA). The eluent and suppressor were 39 mM KOH and
ASRSIIULTRA-4 mm and 20 mM MSA and CSRSIIULTRA-4 mm for anions and cations,
respectively. Finally, concentrations were determined using a conductivity detector (Dionex,
USA), which was calibrated against eight aqueous standards. The detection limit, defined as
3σ, was approximately 0.01–0.09 μg/m$^3$.
Carbonaceous components were measured at the Desert Research Institute (the
thermal/optical reflectance (TOR) method, Reno, NV, USA) using the Interagency
Monitoring of Protected Visual Environments (IMPROVE) TOR protocol. OC comprising
OC1, OC2, OC3, and OC4 was determined at 120℃, 250℃, 450℃, and 550℃, respectively,
in a He atmosphere. EC was analyzed as EC1, EC2, and EC3 at 550℃, 700℃, and 850℃,
respectively, after introducing 2% $O_2$/ 98% He. Pyrolyzed OC (OP) was measured in the



$O_2$/He atmosphere before the reflected light returned to its initial value (Lim et al., 2012;

133   2014).



**2     Measurement overview**

From August 2007 to December 2012, the average $PM_{10}$, $PM_{2.5}$, and $PM_1$ mass concentrations
of all measurements were 30 μg/m$^3$, 19 μg/m$^3$, and 14 μg/m$^3$, respectively (Table 1). The
$PM_{10}$ mass was almost equally partitioned between <1 μm and 1 – 10 μm. Moreover, the
mass of particles between 1 μm and 2.5 μm was considerable and comprises 26% of $PM_{2.5}$
mass. As summarized in Table 1, $SO_4^{2-}$ and OC were the most abundant, followed by $NH_4^+$
and $NO_3^-$. These four species accounted for 48%, 58%, and 69% of the $PM_{10}$, $PM_{2.5}$, and $PM_1$
mass, respectively. Of these species, $SO_4^{2-}$, $NH_4^+$, and EC were pre-dominant in $PM_1$, which
corresponds to more than 75% of those in $PM_{10}$. In comparison, about 65% of OC was
partitioned into $PM_1$. It was even less for $NO_3^-$ as 33%. It is well known that $NO_3^-$ is more
abundant in coarse mode particles due to high affinity to soil mineral. It is also noteworthy
that a substantial amount of OC and EC (20 %) was associated with particles between 1 μm
and 2.5 μm. Of OC sub-components, OC3 and OC4 were mainly associated with coarse
particles. It is evident that $Na^+$ and $Cl^-$ were highly enriched in coarse particles between 2.5
μm and 10 μm.

The high $PM_{10}$ mass was usually observed in the spring along with increased concentrations
of $Ca^{2+}$ and $Mg^{2+}$ (Fig. 1). In comparison, the mass of $PM_1$ was higher in the cold season (fall
to winter) when the concentrations of $SO_4^{2-}$, $NO_3^-$, $NH_4^+$, $K^+$, OC, and EC were highly
elevated. High $PM_{2.5}$ concentrations were observed in both winter and spring periods. Overall,
the three particle masses and their major constituents were highly elevated when dust and
haze events occurred.





Five dust events took place in March, May, November, and December for the entire
experiment period. Upon dust incidence, the daily average mass concentrations of $PM_{10}$ and
$PM_{2.5}$ were enhanced by 3 and 2 times, respectively. In comparison, concentrations of mass,
secondary ions, and carbonaceous compounds were elevated more than two times in $PM_1$
during the haze events from October to April. On March 20, 2010, dust and haze event
occurred concurrently, leading to a maximum $PM_{10}$ concentration of 199 μg/m$^3$. $PM_{10}$
concentrations were occasionally elevated without an official report of KMA on dust
occurrence. For instance, in March 2008, the concentrations of $PM_{10}$ mass, $Ca^{2+}$, and $Mg^{2+}$
were higher than those of dust event days.


**3   PCA analysis of $PM_{10}$, $PM_{2.5}$, and $PM_1$**

Principle component analysis (PCA) was conducted for all measured species of $PM_{10}$, $PM_{2.5}$,
and $PM_1$ aerosols including water-soluble ions, OC, EC, and mass for the whole period. PCA
analysis identifies the correlation of variables through orthogonal transformation and
summarizes the main characteristics of measurement data set. In PCA analysis, two
components are usually selected. In this study, the principle component 1 and 2 accounted for
more than 60% of the total variance of $PM_{10}$, $PM_{2.5}$, and $PM_1$. The principle component 1
(PC1) was composed of high loadings for $SO_4^{2-}$, $NO_3^-$, $NH_4^+$, $K^+$, OC, and EC, especially in
$PM_1$ (Fig. 2). These six species contributed almost equally to the PC1, which explained 46%
of the total variance. In contrast, the principle component 2 (PC2) explained 16% of the total
variance and was characterized by high loadings for $Na^+$, $Cl^-$, $Mg^{2+}$ and $Ca^{2+}$, mainly in $PM_{10}$
and $PM_{2.5}$. Interestingly, the principle component 3 (PC3) comprising 9% of the total variance,
was associated with high loadings for $NH_4^+$ and $Ca^{2+}$, particularly in $PM_{10}$ (Fig. 2 and 3).
These three independent factors explain more than 70% of the total variance.

As the most abundant species, $SO_4^{2-}$ and $NO_3^-$ concentrations were highly correlated with
PC1 loadings for all three-size particles (Fig. 3), confirming that the PC1 represents the
influence of anthropogenic pollution sources (Zhang et al., 2013b). So were OC2 and EC1



that have been reported to originate from biomass combustion sources (e.g., Lim et al., 2012).
In PC2, the loadings for $Ca^{2+}$, $Mg^{2+}$, $Na^+$, and $Cl^-$ were the highest and well correlated with
OC4 concentration in $PM_{2.5}$ and $PM_{10}$, which used to be elevated upon dust events (Lim et al.,
2012). In saline dust, the concentrations of $Ca^{2+}$, $Na^+$, and $Cl^-$ were enhanced concurrently
with OC sub-component (Zhang et al., 2014; Shang et al., 2018; O'Dowd et al., 2004; Griffith
et al., 2010). The sea-salt contribution of $Ca^{2+}$ was estimated to be 12% in $PM_{2.5}$ and 19% in
$PM_{10}$, assuming that sodium was derived solely from sea salt. In this study, the measurements
of water-soluble ions demonstrate that the contribution of sea salt species was found to reach
the maximum in summer when aerosol loading is at its minimum under influence of marine
air. Thus, the PC2 represents the impact of dust particles including alkaline soils. $NH_4^+$
concentration was moderately related to PC3 loadings in $PM_{2.5}$ and $PM_{10}$. In particular, a
relatively good correlation of $NH_4^+$ with $Ca^{2+}$ in $PM_{10}$ indicates the agricultural influence due
to fertilizer use. It is noteworthy that PC3 loading was high in spring and summer when the
concentrations of particulate matter were low with reduced continental outflows. In addition,
the PC3 loadings increased with time, reaching to the highest in 2010. The recent studies also
reported that in China, $NH_3$ emission was increased due to fertilizer application and $NH_4^+$
concentration was higher in spring and summer than the other seasons (Warner et al, 2017;
Kang et al., 2016).

Therefore, the three principle components manifest the main sources of particulate matters in
the study region. As anthropogenic sources, PC1 is predominant in $PM_1$ and $PM_{2.5}$. PC2
demonstrates the influence of soil dust on $PM_{10}$ and $PM_{2.5}$. Fertilizer use is likely responsible
for the variance of PC3. In order to estimate the contribution of these three factors to the mass
of $PM_{10}$, $PM_{2.5}$, and $PM_1$ at GCO, multi-linear regression analysis was conducted using factor
loadings, of which result is given below:

$PM_{10} \ (\mu g/m^3) = 31.1 + 4.7 \, PC1 + 3.7 \, PC2 + 4.1 \, PC3 \quad (r = 0.89, P = 0.03)$ \hfill (1)
$PM_{2.5} \ (\mu g/m^3) = 19.2 + 3.1 \, PC1 + 0.4 \, PC2 + 1.9 \, PC3 \quad (r = 0.95, P = 0.03)$ \hfill (2)
$PM_1 \ \ (\mu g/m^3) = 14.8 + 2.5 \, PC1 - 0.7 \, PC2 + 1.6 \, PC3 \quad (r = 0.93, P = 0.04)$ \hfill (3)
, where PC1, PC2, and PC3 are factor loadings.






The intercepts of these three equations are equivalent to the average concentrations for $PM_{10}$,
$PM_{2.5}$, and $PM_1$ (Table 1 and 2). It confirms that the three PCs are sufficient enough to
explain the variation of aerosol masses observed at GCO. It is evident that PC1 is a dominant
factor determining the particulate mass of $PM_{10}$ (63%) as well as $PM_1$ (99%) and $PM_{2.5}$ (90%).
PC2 was most evident in $PM_{10}$ (36%) and not negligible in $PM_{2.5}$ (9%). It is worthy
emphasizing that $NH_4^+$ factor was distinguished as PC3, even though its contribution was the
least. In addition, the very small or negative loading of PC2 for $PM_{2.5}$ and $PM_1$ suggests the
scavenging of anthropogenic pollutants on dust particles.

**4    Diagnosis of dust and haze**

While soil dust has been recognized as a main driver for high $PM_{10}$ mass in northeast Asia
(Yang et al., 2009), air pollution events are typically distinguished by the concentrations of
$PM_{2.5}$ (EPA, 2012). In Korea, the aerosol mass concentrations have been often elevated upon
Asian dust or haze occurrence. In this context, mode analysis of $PM_{10}$, $PM_{2.5}$, and $PM_1$ mass
concentrations was conducted to diagnose the impact of dust and haze particles on particulate
matter. The frequency distributions of all $PM_{10}$, $PM_{2.5}$, and $PM_1$ measurements are shown in
Figure 4. For the three-size aerosol masses, the main-mode concentrations are comparable to
the median concentrations (Table 3 and Fig. 4). The main-mode concentration of $PM_{10}$ and
$PM_{2.5}$ was 25 $\mu g/m^3$ and 16 $\mu g/m^3$, respectively, which are much lower than those of national
standard of annual mean of 50 $\mu g/m^3$ and 25 $\mu g/m^3$, respectively. The main-mode
concentration of $PM_1$ (11 $\mu g/m^3$) was similar to the air quality guideline of the World Health
Organization (WHO) for $PM_{2.5}$ (10 $\mu g/m^3$) (WHO, 2006). Of $PM_{2.5}$ mass, the contribution of
mineral dust was estimated to be ~10% in previous section, which is equivalent to about 2
$\mu g/m^3$.

The mean concentrations of the three types of particulate matters were higher than their
median and main-mode concentrations and the standard deviations were comparable to the
median concentrations. These results show that mass concentrations varied in a wide range





due to high concentration events. For $PM_{10}$, the mean+σ of 52 μg/m$^3$ was close to the national
standard of $PM_{10}$ annual average concentration. While the mean+σ concentration of $PM_{2.5}$
was higher by 28% than the national standard of 25 μg/m$^3$, the mean+σ of $PM_1$ (25 μg/m$^3$)
met the annual standard of $PM_{2.5}$ concentration. The mean+σ concentrations of $PM_{10}$, $PM_{2.5}$,
and $PM_1$ were commensurate with the 90$^{th}$ percentiles that generally represent the highest
concentration of the long-tern measurements.

In Korea, dust occurrence is determined by eye observation and haze is recorded when RH is
less than 75 % and visibility is between 1 km and 10 km. The concentrations of individual
dust and haze samples are presented in the bottom of Figure 4. While the $PM_{10}$ and $PM_{2.5}$
concentrations of five dust events are placed in the range above the mean+σ, all of the high
$PM_{10}$ concentrations were not observed on dust days. In contrast, the concentrations of all
haze samples were over the mean+σ of $PM_1$ (Fig. 4).

When $PM_{10}$ and $PM_{2.5}$ mass belong to the top 10%, their $Ca^{2+}$ and $Mg^{2+}$ concentrations were
also within the highest 10 % of the entire measurements. Particularly, $Ca^{2+}$ concentration (0.7
μg/m$^3$) was 3 times as high as the average concentration for both $PM_{10}$ and $PM_{2.5}$ (Table 1),
implying that dust effect is not negligible in $PM_{2.5}$.

At GCO, the five-year measurements of aerosol mass and chemical composition reveal that
the top 10 % of $PM_{10}$, $PM_{2.5}$, and $PM_1$ mass was affected by dust or haze plumes and their
effect is traceable by the 90$^{th}$ percentile mass concentrations of particulate matter. If $PM_{10}$ or
$PM_{2.5}$ mass concentrations are above the 90$^{th}$ percentile, airborne dust particles played a
substantial role in mass enhancement, regardless of the occurrence of event. Likewise,
anthropogenic pollution is a main driver for enhanced $PM_1$ and $PM_{2.5}$ mass concentrations if
their concentrations are greater than the 90$^{th}$ percentile. For $PM_{2.5}$, the impact of mineral dust
should be considered in northeast Asia region downwind of the dust belt.




## 5    Conclusions

At GCO, filter samples for $PM_1$, $PM_{2.5}$, and $PM_{10}$ were collected and their mass, water-soluble inorganic ions and carbonaceous compounds were analyzed from 2007 to 2012. For the entire period, the average concentrations of $PM_{10}$, $PM_{2.5}$, and $PM_1$ were 30, 19, and 14 $\mu g/m^3$, respectively. $PM_{2.5}$ accounted for 63% of $PM_{10}$, while $PM_1$ comprised 74% of $PM_{2.5}$ on average.

From the principle component analysis using all measured species for $PM_{10}$, $PM_{2.5}$, and $PM_1$, the three principle components (PC1, PC2, and PC3) were distinguished, which explained 46%, 16%, and 9% of the total variances, respectively. The PC1 representing the effect of anthropogenic pollution was characterized by high loadings of $SO_4^{2-}$, $NO_3^-$, $NH_4^+$, $K^+$, OC, and EC. The PC2 was distinct with high loadings for $Ca^{2+}$ and $Mg^{2+}$ that originate from soil dust. Although the contribution was low, the PC3 was significant for two reasons. First, the loadings of PC3 showed an increasing tendency over time. In addition, the PC3 loadings were discernible during warm season, in contrast to other two components that explain the variations of mass and major constituents of aerosol during cold season. The multiple regression using the three PC loadings shows that the anthropogenic pollution accounted for 99 % and 63 % of $PM_1$ and $PM_{10}$ mass variation, respectively. The effect of soil dust was the largest on $PM_{10}$ (36%) and not negligible on $PM_{2.5}$ (~10%).

The mode analysis of $PM_{10}$, $PM_{2.5}$, and $PM_1$ mass concentrations demonstrates that the main mode was commensurate with the median concentration and the mean + $\sigma$ was comparable to the concentration of the 90[th] percentile. It indicates that the average mass concentration is highly susceptible to high-concentration episodes. Consequently, the mean+$\sigma$ is suggested as a robust criterion that determines the substantial impact of soil dust or pollution plumes on $PM_{10}$, $PM_{2.5}$, and $PM_1$. Furthermore, the results of this study reveal that in northeast Asia, non-combustion sources such as soil dust with impose constraints to the reduction of $PM_{2.5}$ as well as $PM_{10}$ concentrations and raise questions about the efficacy of yearly average concentrations as environmental standards.



**Acknowledgments**

This research was supported by the National Strategic Project-Fine Particle of the National Research Foundation of Korea (NRF) funded by the Ministry of Science and ICT (MSIT), the Ministry of Environment (ME), and the Ministry of Health and Welfare (MOHW) (2017M3D8A1092015). We also thank the Korea Meteorological Administration for supplying event information including dust and haze, the National Institute of Environmental Research, Gwangju Institute of Science and Technology, and Seoul National University for supporting the experimental data.

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



Table 1. Mean and mean + σ (standard deviation) concentrations [µg/m³] of mass and
chemical constituents for $PM_{10}$, $PM_{2.5}$, and $PM_1$ at GCO during 2007 ~ 2012.

| 499 | $PM_{10}$ | | $PM_{2.5}$ | | $PM_1$ | |
|---|---|---|---|---|---|---|
| | Mean | Mean+σ | Mean | Mean+σ | Mean | Mean+σ |
| Mass | 30 | 52 | 19 | 32 | 14 | 25 |
| $Cl^-$ | 0.8 | 1.7 | 0.1 | 0.3 | 0.1 | 0.2 |
| $NO_3^-$ | 2.1 | 4.1 | 1.0 | 2.2 | 0.7 | 1.9 |
| $SO_4^{2-}$ | 5.5 | 9.5 | 4.4 | 7.5 | 4.3 | 7.5 |
| $Na^+$ | 1.2 | 1.9 | 0.4 | 0.7 | 0.2 | 0.4 |
| $NH_4^+$ | 2.8 | 4.6 | 2.4 | 4.0 | 2.1 | 3.6 |
| $K^+$ | 0.3 | 0.5 | 0.2 | 0.4 | 0.2 | 0.4 |
| $Mg^{2+}$ | 0.2 | 0.3 | 0.1 | 0.1 | 0.02 | 0.04 |
| $Ca^{2+}$ | 0.3 | 0.7 | 0.1 | 0.3 | 0.1 | 0.2 |
| OC | 4.0 | 6.6 | 3.4 | 5.7 | 2.6 | 4.3 |
| OC1 | 0.1 | 0.2 | 0.1 | 0.2 | 0.1 | 0.2 |
| OC2 | 0.8 | 1.3 | 0.8 | 1.3 | 0.7 | 1.1 |
| OC3 | 1.2 | 2.0 | 0.9 | 1.5 | 0.7 | 1.1 |
| OC4 | 0.9 | 1.7 | 0.7 | 1.3 | 0.4 | 0.8 |
| OP | 0.9 | 1.7 | 0.9 | 1.6 | 0.7 | 1.3 |
| EC | 1.5 | 2.9 | 1.5 | 2.7 | 1.2 | 2.0 |
| EC1 | 1.2 | 2.5 | 1.1 | 2.3 | 0.8 | 1.5 |
| EC2+EC3 | 0.3 | 0.5 | 0.4 | 0.6 | 0.4 | 0.6 |





Table 2. Intercepts and coefficients for multi-linear regression of $PM_{10}$, $PM_{2.5}$, and $PM_1$ mass

501         concentrations using the three principle components (PC1, PC2, and PC3).


|  | $PM_{10}$ | $PM_{2.5}$ | $PM_1$ |
|---|---|---|---|
| PC1 | 4.7 | 3.1 | 2.5 |
| PC2 | 3.7 | 0.4 | −0.7 |
| PC3 | 4.1 | 1.9 | 1.6 |
| Intercept | 31.1 | 19.2 | 14.8 |





Table 3. The statistical summary for mass concentrations of $PM_{10}$, $PM_{2.5}$, and $PM_1$ over entire

504        experiment period [$\mu g/m^3$].

|           | Median | Mean | S. D. ($\sigma$) | Main mode | Mean+$\sigma$ |
|-----------|--------|------|------------------|-----------|---------------|
| $PM_{10}$  | 24     | 30   | 22               | 25        | 52            |
| $PM_{2.5}$ | 15     | 19   | 13               | 16        | 32            |
| $PM_1$     | 11     | 14   | 11               | 11        | 25            |







507 Figure 1. Time-series variations of major constituents of PM$_{10}$, PM$_{2.5}$, and PM$_1$ for the entire

508   experiment [μg/m$^3$]. Spring and winter periods are shaded in orange and gray.





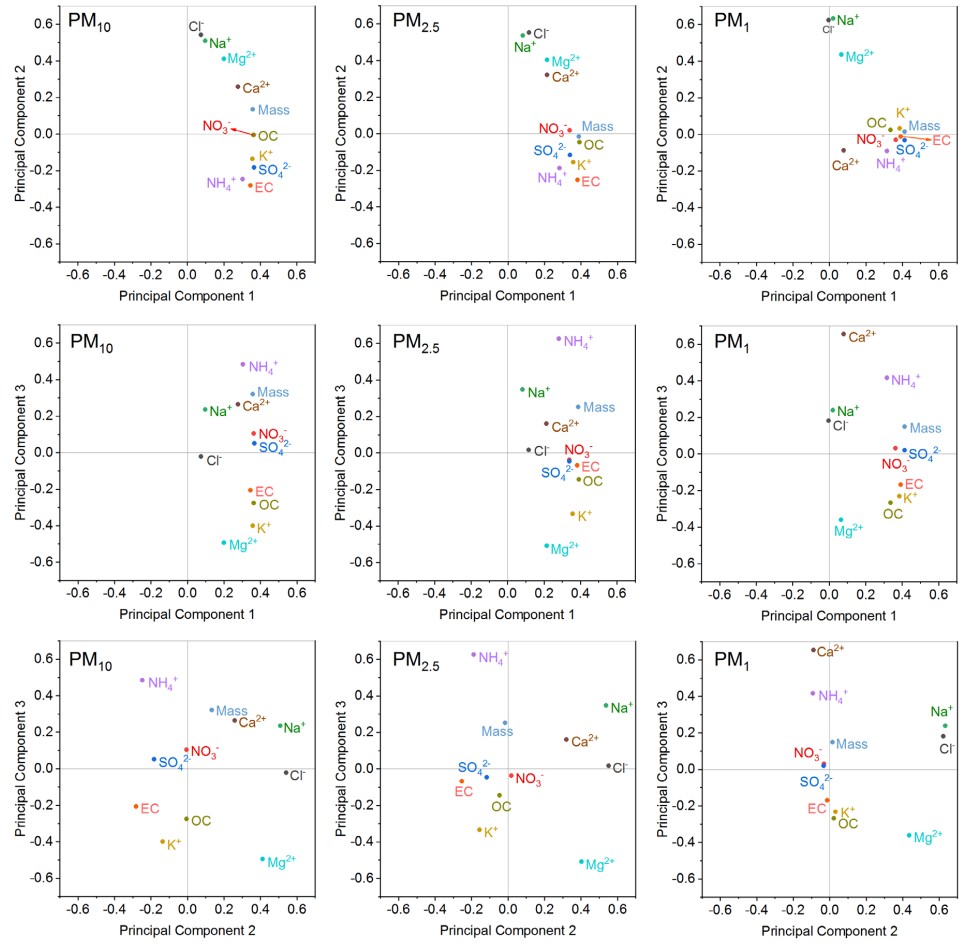


Figure 2. The results of Principal Component Analysis of all measured species including mass,

511           water-soluble ions, OC, and EC for $PM_{10}$, $PM_{2.5}$, and $PM_1$.





Figure 3. Correlations between the three principle component loadings and major species

514          concentrations [μg/m³] for $PM_{10}$, $PM_{2.5}$, and $PM_1$.



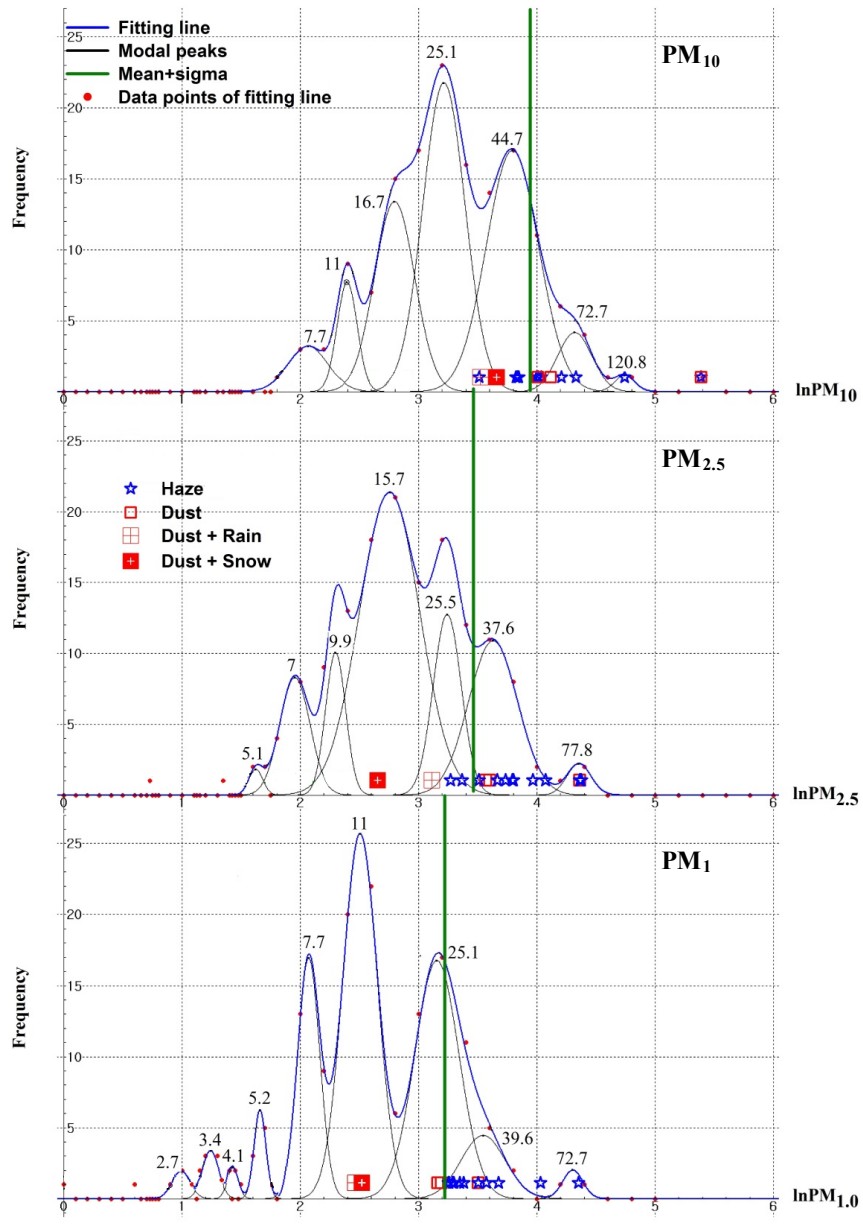


Figure 4. Frequency distributions of $PM_{10}$, $PM_{2.5}$ and $PM_1$ mass concentrations for all
measurements. Mass concentrations are given as ln values in x-axis. The green
lines stand for mean+σ. The individual samples collected during dust or haze
events are marked as different symbols along the x-axis.