# Peer review of "Diagnosis of dust- and haze pollution-impacted PM$_{10}$, PM$_{2.5}$,"

_Atmospheric Chemistry and Physics, 2018_

## Referee Comment (RC1) · Anonymous Referee #2 · 2 Oct 2018

This work reported five-year measurements of the mass concentration and composition of PM10, PM2.5 and PM1 at Gosan climate observatory. Based on the concentration of soluble ions and OC/EC data, PCA and distribution frequency analysis were performed to understand the influence of soil dust and anthropogenic pollution on bulk aerosol. Although the authors provided long term dataset of PM at this observation site, the results are not so attractive based on their analysis. At the current state, it is not publishable on ACP. 1. As for principle component analysis, three factors explained 71% of the total variance. It means 29% of the variance was not explained. The residue was quite large. What's the possible contribution for this unexplained part? 2. Because the number of the tracer was limited and the tracers were lack of uniqueness,

the results about source apportionment were not so robust. For example, PC3 was dominated with NH4+ and Ca2+, while secondary formation of NH4+ might also lead to the correlation between NH4+ and Ca2+. 3. The time series of mass concentration for each PC should be given after the PCA data. This is important for understanding the sources of PM. 4. It was unclear about the diagnosis of dust and haze based on frequency distribution. There was no robust criterion to differentiate haze from dust events. As shown in Fig.4, the mass loading of PM on dust or haze events varied greatly. The data could not well support the conclusions drawn by the authors.

---

## Referee Comment (RC2) · Tao (Referee) · 28 Nov 2018

This is an article about Diagnosis of dust- and haze pollution-impacted PM10, PM2.5, and PM1 aerosols observed at Gosan Climate Observatory. I have some comments and suggestions.

-Introduction, this section didn't write about the main content such as pollution characteristic of water ions, OC and EC in PM, method of source apportionment. -Line 105-106, please add the map of sampling site. -Line 104-108, how about temporal distribution of the samples or how many samples in each season per year? -Line 109-110, what are the standards of dust event and haze event? -Line115-116, what is

the method or condition of extracted water-soluble ions? How about extraction times? How to ensure the extraction efficiency if only one extraction? -Line 144, "of these species...and EC were pre-dominant....," from Table 1, EC maybe OC. -Line145-146, "In comparison, about 65% of OC was partitioned into PM1. It was even less for NO3- as 33%". These two sentences are not clear. -Table 1, Please add minimum, maximum, median, percentile(10th, 90th) inTable 1 and revise mean+$\sigma$ to mean±SD. To revise title of Table 1. -Please add figure to show the species proportion in PM. -Line 153-168, the describe for figure 2 is not accurate. -PCA analysis of PM, two components are usually selected, but it uses three components in multi-linear regression analysis, why? I don't think it is right to using mass of PM as factor in PCA analysis because mass of PM acts as dependent variable in multi-linear regression analysis. -I think it exists errors if the study didn't exclude dust and haze event in PCA analysis. -Perhaps you maybe use PMF model to analyze source of PM. -Line 49-52, Please add the instrument condition of analyzing ions. How about pretreatment of PM samplers for ions analysis and how to quantify the concentrations of ions? -what is basis of dust and haze diagnosis based on method using in this study. -Conclusions, this section is too long. -Table 1 and 3 can be merged. -Figure 1, There are no data of winter in 2009 and spring in 2011 and 2012. It is scare of persuasion based on Figure 1 about spring.

---

## Author Comment (AC1) · 25 Dec 2018

Correspondence to Referee #2

Thank you very much for your thorough and constructive comments on our manuscript acp-2018-721, entitled "Diagnosis of dust- and haze pollution-impacted PM10, PM2.5, and PM1 aerosols observed at Gosan Climate Observatory". We added all available information to provide more solid evidences and revised the manuscript according to your comments. The response is given to each comment. In the revised manuscript, changes are colored in blue and page and line numbers are given for the revised manuscript.

[Figure]

General Comments

Comment 1: As for principle component analysis, three factors explained 71% of the total variance. It means 29% of the variance was not explained. The residue was quite large. What's the possible contribution for this unexplained part? Response 1: As you pointed out, the residual is about 30%. In general, the two factors of the largest distance in eigenvalue are chosen for PCA, which account for more than 60% of the total variance in this study (Figure given below). The reason for we selected three principle components is not only that the 3rd factor (PC3) is in charge of 9 % of variance, but also that PC3 exhibits a clear increasing trend with time (for details of increasing trend see Response 2). In addition, PC3 loadings were discernible during warm season, in contrast to other two components that were dominated in cold season. The contribution of the rest components was much less than 10 %. For instance, the fourth component (PC4) accounts for 6 % and indicates salts influence by high loadings of $Na_+$ and $Cl_-$. The rest components contributed no larger than 5 %. This information is added to Supplementary Information (SI 3).

SI 3. Eigenvalues according to the number of principal components.

Comment 2: Because the number of the tracer was limited and the tracers were lack of uniqueness, the results about source apportionment were not so robust. For example, PC3 was dominated with $NH_4^+$ and $Ca_{2+}$, while secondary formation of $NH_4^+$ might also lead to the correlation between $NH_4^+$ and $Ca_{2+}$. Response 2: There is no fancy item in the measurement list. Nevertheless, you agreed to review this manuscript and gave constructive comment. We really appreciate it. In order to understand the factors determining the variation and magnitude of particulate matters in the study region, we had measured major aerosol constituents in three size-cuts for a long period. This is we believe the uniqueness of this study that led us to main findings.

$NH_4^+$ concentration used to be in good correlation with PM mass, $SO_4^{2-}$ or $NO_3^-$. $NH_4^+$ is a neutralizing compound of aerosol and is so $Ca_{2+}$, which is an indicator

for the impact of Asian dust in the study region. $NH_3$ concentration is the highest due to fertilizer application in spring and summer, and enhanced due to anthropogenic emissions in winter. On the other hand, the source strength of $Ca_{2+}$ is the greatest in spring because of the sandstorm transported by northerly and northwesterly wind. $NH_4^+$ is possibly correlated with $Ca_{2+}$ due to secondary formation but in association with $SO_4^{2-}$ or $NO_3^-$.

In previous studies, Sulfur Oxidation Rate (SOR) = $[nSO_4^{2-}/(nSO_4^{2-}+nSO_2)]$ and Nitrogen Oxidation Rate (NOR) = $[nNO_3^-/(nNO_3^-+nNO_2)]$ was found to be high during summer (n represents molar concentration), which indicates the efficient conversion of precursor gases to secondary ions in particles. In this study, the SOR and NOR were not increased with time like PC3 loadings presented in the following figure ($R^2$ = 0.53), implying that there was no significant relation between PC3 and secondary formation. To avoid the confusion, therefore, the relevant discussion was added in the revised manuscript.

Time series of SOR, NOR, PC3 scores, and linear fitting of PC3.

Page 9 line 236-237: "These results indicated the agricultural influence on PC3 loading due to fertilizer usage."

Comment 3: The time series of mass concentration for each PC should be given after the PCA data. This is important for understanding the sources of PM. Response 3: According to your suggestion, we provide the time series plot of PC loadings to the Supplementary Information.

SI 4. Time series of scores for each principal component.

High scores of PC1 along with the high $SO_4^{2-}$ and $NO_3^-$ concentrations were observed particularly in winter, confirming that PC1 represented the influence of anthropogenic pollution sources (Zhang et al., 2013b). In comparison, PC2 loading was high for $Ca_{2+}$, $Mg_{2+}$, $Na_+$, and $Cl_-$ in spring and well correlated with OC4 concentration in

[Figure]

PM2.5 and PM10, which used to be elevated upon dust events (Lim et al., 2012). It is noteworthy that PC3 loading was high in spring and summer and increased with time, reaching to the highest in 2010. In addition, NH4+ concentration was moderately related to PC3 loadings in PM2.5 and PM10 with low concentrations of particulate matter, indicating the agricultural influence on PC3 loading due to fertilizer usage.

Comment 4: 4. It was unclear about the diagnosis of dust and haze based on frequency distribution. There was no robust criterion to differentiate haze from dust events. As shown in Fig.4, the mass loading of PM on dust or haze events varied greatly. Response 4: First of all, it is not the purpose of this study to differentiate between haze and dust. In Korea, the reference concentrations of PM10 and PM2.5 have been established for warning or air quality forecast. However, there are no criteria for mass concentrations of haze or dust occurrence, which was the motivation of this study. The result of this study reveals that the GCO PMs are under consistent influence of dust, haze, or the two being mixed. The samples of PM10, PM2.5 and PM1 were concurrently collected approximately every 6–8 days during the five years. As a result, our samples do not cover all officially issued haze and dust events, thereby being suitable for diagnosing the effect of haze and dust on particulate matters.

We also tested if these criteria were valid for the recent measurements at Gosan from January 2016 to October 2017. PM2.5 measurement officially began in 2015 and the hourly measurements of PM10 and PM2.5 are available through http://www.airkorea.or.kr/. During this period, dust and haze events were observed for eight and twelve days, respectively, for which daily average concentrations are presented in Figure 5. As shown in Figure 5, all dust and haze events are found above the mean+SD of PM10 (52 $\mu$g/m3) and PM2.5 (32 $\mu$g/m3), respectively, even for the recent two years. It demonstrates that the criteria suggested in this study are robust and useful to diagnose the effect of dust and haze impacted particles.

Figure 5 is replaced with the one shown below in the revised manuscript.

Figure 5. Frequency distributions of PM10, PM2.5 and PM1 mass concentrations for all measurements. Mass concentrations are given as ln values in x-axis. The green lines stand for mean+SD. The individual samples collected during dust or haze events are marked as different symbols along the x-axis. For comparison, added right below the x-axis are the daily average concentrations of haze and dust days during January 2007 ∼ October 2012 (http://www.airkorea.or.kr/).

All figures are in the file uploaded in Supplement.

Please also note the supplement to this comment:
https://www.atmos-chem-phys-discuss.net/acp-2018-721/acp-2018-721-AC1-supplement.pdf
* * *
Eigenvalues

15

10

0

0    5    10   15   20   25   30   35

Principal Component Number

**Fig. 1.**

[Figure]

**Fig. 2.**

[Figure]

**Fig. 3.**

[Figure]

**Fig. 4.**

---

## Author Comment (AC2) · 25 Dec 2018

Correspondence to Referee Yan Tao

Thank you very much for your thorough and constructive comments on our manuscript acp-2018-721, entitled "Diagnosis of dust- and haze pollution-impacted PM10, PM2.5, and PM1 aerosols observed at Gosan Climate Observatory". We added all available information to provide more solid evidences and revised the manuscript according to your comments. The response is given to each comment. In the revised manuscript, changes are colored in blue and page and line numbers are given for the revised manuscript.

[Figure]

Comment 1: Introduction, this section didn't write about the main content such as pollution characteristic of water ions, OC and EC in PM, method of source apportionment.
Response 1: The statements regarding pollution and haze particles and method of source apportionment were added to the revised manuscript (Page 4 line 75-89, Page 5 line 105-111, Page 7 line 161-164 ).

Page 4 line 75-89: "The Northeast Asia is the region with the highest sulfur and nitrogen concentrations and deposition due to large emissions of SO2 and NOx (Vet et al., 2014). Particularly in China, the highest concentration of SO42- was observed in the period of 2000-2004. The anthropogenic SO2 emission of China accounted for about one-fourth of the world and 90 % of the East Asia emission since the 1990's (Ohara et al., 2007). However, the increasing rate of SO2 emission has slowed down and decreased after 2006 due to series of policy implements for reducing fine aerosols (van der A et al., 2017). In contrast, NOx and NO3- concentrations have been highly increased in megacities of China during severe pollution episodes (e.g. Shang et al., 2018c). The increase in bulk nitrogen deposition was not only driven by increased NOx concentration but also NH3 enhancement over the time (Liu et al., 2013). In addition, VOCs emission has been continuously increased in China (Hong et al., 2017). The largest source of NH3 and VOCs was found in India and China (Behera et al., 2013; Xu et al., 2018). While being transported away from China, these pollutants were often mixed with dust particles and raised the PM2.5 concentration in Korea. In this case, it is difficult to identify the main cause of air quality deterioration." Page 5 line 105-111: "As a method of source apportionment, factor analysis successfully extracted dust impact on PM10 by high loadings of Mg2+ and Ca2+ as well as crustal elements (e.g., Choi et al., 2001). Heavy metals have been used to identify various types of urban dust in Central China using principle component analysis (PCA) (e.g., Han et al., 2006). For large sets of measurements, positive matrix factorization (PMF) is a powerful tool to quantitatively distinguish different types of sources (e.g. Gupto et al., 2012). With less parameters, non-negative matrix factorization was also useful to identify major sources of aerosols (e.g., Shang et al., 2018b)."

Page 7 line 161-164: "To understand the factors determining the variation of particulate matters and diagnose the influence of dust and pollution on them, PCA analysis was performed using long-term measurements of PM10, PM2.5, and PM1 mass and major chemical constituents including eight water-soluble ions, OC, and EC."

Behera, S. N., Sharma, M., Aneja, V. P., and Balasubramanian, R.: Ammonia in the atmosphere: a review on emission sources, atmospheric chemistry and deposition on terrestrial bodies, Environ. Sci. Pollut. R., 20, 8092–8131, https://doi.org/10.1007/s11356-013-2051-9, 2013.

Hong, C., Zhang, Q., He, K., Guan, D., Li, M., Liu, F., and Zheng, B.: Variations of China's emission estimates: response to uncertainties in energy statistics, Atmos. Chem. Phys., 17, 1227–1239, https://doi.org/10.5194/acp-17-1227-2017, 2017.

Liu, X. J., Zhang, Y., Han, W., Tang, A., Shen, T., Cui, Z., Vitousek, P., Erisman, J. W., Goulding, K., Christie, P., Fangmeier, A., and Zhang, F.: Enhanced nitrogen deposition over China, Nature, 494, 459–462, https://doi.org/10.1038/nature11917, 2013.

Lu, Z., Streets, D. G., Zhang, Q., Wang, S., Carmichael, G. R., Cheng, Y. F., Wei, C., Chin, M., Diehl, T., and Tan, Q.: Sulfur dioxide emissions in China and sulfur trends in East Asia since 2000, Atmos. Chem. Phys., 10, 6311–6331, https://doi.org/10.5194/acp-10-6311-2010, 2010.

Ohara, T., Akimoto, H., Kurokawa, J., Horii, N., Yamaji, K., Yan, X., and Hayasaka, T.: An Asian emission inventory of anthropogenic emission sources for the period 1980–2020, Atmos. Chem. Phys., 7, 4419–4444, https://doi.org/10.5194/acp-7-4419-2007, 2007.

Shang, X., Zhang, K., Meng, F., Wang, S., Lee, M., Suh, I., Kim, D., Jeon, K., Park, H., Wang, X., and Zhao, Y.: Characteristics and source apportionment of fine haze aerosol in Beijing during the winter of 2013, Atmos. Chem. Phys., 18, 2573–2584, https://doi.org/10.5194/acp-18-2573-2018, 2018b.

van der A, R. J., Mijling, B., Ding, J., Koukouli, M. E., Liu, F., Li, Q., Mao, H., and Theys, N.: Cleaning up the air: effectiveness of air quality policy for SO2 and NOx emissions in China, Atmos. Chem. Phys., 17, 1775–1789, https://doi.org/10.5194/acp-17-1775-2017, 2017.

Vet, R., Artz, R.S., Carou, S., Shaw, M., Ro, C.U., Aas, W., Baker, A., Bowersox, V.C., Dentener, F., Galy, L.C., Hou, A., Pienaar, J.J., Gillett, R., Forti, M.C., Gromov, S., Hara, H., Khodzherm, T., Mahowald, N.M., Nickovic, S., Rao, P.S.P., Reid, N.W.,: A global assessment of precipitation chemistry and deposition of sulfur, nitrogen, sea salt, base cations, organic acids, acidity and pH and phosphorus. Atmos. Environ. 93, 3-100, 2014.

Xu, R., Tian, H., Pan, S., Prior, S. A., Feng, Y., Batchelor, W. D., Chen, J., and Yang, J.: Global ammonia emissions from synthetic nitrogen fertilizer applications in agricultural systems: Empirical and process-based estimates and uncertainty, Glob. Chang Biol., 25, 314–326, https://doi.org/10.1111/gcb.14499, 2018.

Choi, J. C., Lee, M., Chun, Y., Kim, J., and Oh, S.: Chemical composition and source signature of spring aerosol in Seoul, Korea, J. Geophys. Res. Atmos., 106, 18067–18074, https://doi.org/10.1029/2001JD900090, 2001.

Gupta, A. K., Karar, K., and Srivastava, A.: Chemical mass balance source apportionment of PM10 and TSP in residential and industrial sites of an urban region of Kolkata, India, J. Hazard. Mater. 142, 279–287, https://doi.org/10.1016/j.jhazmat.2006.08.013, 2007.

Han, Y. M., Du, P. X., Cao, J. J., and Posmentier, E. S.: Multivariate analysis of heavy metal contamination in urban dusts of Xi'an, Central China, Sci. Total Environ., 355, 176–186, https://doi.org/10.1016/j.scitotenv.2005.02.026, 2006.

Comment 2: Line 105-106, please add the map of sampling site. Response 2: It was not given in the submitted manuscript because it has been shown in previous studies.

However, the map of GCO is now in the revised manuscript as Figure 1 with a statement given below.

Page 5 line 124-130: "Aerosol samples were collected separately for PM1, PM2.5, and PM10 onto 37 mm Teflon and Quartz filters (Pall, Corp.) using sharp-cut cyclones (URG, USA) at the Gosan Climate Observatory (GCO) ($33°17′N$, $126°10′E$) from 2007 to 2012 (Fig. 1). As an ideal location to observe continental outflows in northeast Asia, the GCO has been used as a key measurement site not only for intensive field campaigns such as ABC-EAREX2005 (Atmospheric Brown Cloud–East Asia Regional Experiment) (e.g., Lee, et al., 2007), but also long-term studies (e.g., Lim et al., 2018; Shang et al., 2018a)."

Figure 1. The map showing the Gosan Climate Observatory (GCO) site in the western-most part of Jeju Island, South Korea. Comment 3: Line 104-108, how about temporal distribution of the samples or how many samples in each season per year? Response 3: The number of samples per season and year is given in Supplementary information (SI 1) and statement is added to the revised manuscript.

SI 1. The number of samples taken from 2007 to 2012. Season 2007 2008 2009 2010 2011 2012 Spring (Mar.-May) NA* 16 18 10 7 NA* Summer (Jun.-Aug.) 1 3 8 9 4 NA* Fall (Sep.-Nov.) 9 2 11 8 10 NA* Winter (Dec.-Feb.) 2 8 3 10 10 3 *NA is non-available

Page 6 line 135-138: "The samples of PM10, PM2.5 and PM1 were concurrently collected approximately every 6–8 days during the five years. As a result, our samples do not cover all officially issued haze and dust events, thereby being suitable for diagnosing the effect of haze and dust on particulate matters."

Comment 4: Line 109-110, what are the standards of dust event and haze event? Response 4: It was stated in Line 257-258 of the submitted manuscript and Page 11 line 287-288 in the revised manuscript.

Page 11 line 287-288: "In Korea, dust occurrence is determined by eye observation

and haze is issued when RH is less than 75 % and visibility is between 1 km and 10 km." Comment 5: Line115-116, what is the method or condition of extracted water-soluble ions? How about extraction times? How to ensure the extraction efficiency if only one extraction? Response 5: Extraction was done for 20 min by sonication. We tested different extraction times and found that 15~20 minutes were just right. In Figure below, the three-extraction time of 10, 20, and 40 minutes were compared for Cl-, SO42-, and NO3-.

Parallel tests of anion peak height in IC analysis with different sonication time.

The relevant part was modified as follows. Page 6 line 143-144: "Water-soluble species were extracted from the filters into a solution comprising a mixture of 19 mL distilled water and 1 mL methanol via $20-$min sonication."

Comment 6: Line 144, "of these species. . .and EC were pre-dominant. . ..," from Table 1, EC maybe OC. Response 6: It is corrected.

Page 7 line 175: "Of these species, SO42$-$, NH4+, and OC were pre-dominant in PM1"

Comment 7: Line145-146, "In comparison, about 65% of OC was partitioned into PM1. It was even less for NO3- as 33%". These two sentences are not clear. Response 7: It means that PM1_OC/PM10_OC = 65 % and PM1_NO3$-$/PM10_ NO3$-$ = 33 %, respectively. These two sentences were modified as follows.

Page 7 line 176-177: "In comparison, about 65% of the OC in PM10 was partitioned into PM1. It was even less for NO3$-$ at 33%."

Comment 8: Table 1, Please add minimum, maximum, median, percentile(10th, 90th) inTable 1 and revise mean+$\sigma$ to mean$\pm$SD. To revise title of Table 1. Response 8: Table 1 is modified as follows. Also, 'mean+$\sigma$' was changed to 'mean+SD' in all relevant parts including Table 1. For long-term measurements, 10th and 90th percentiles well represent the minimum and maximum concentration and thus, they are presented with

the median.

  Table 1. Statistics* of mass and chemical constituents concentrations [$\mu$g/m3] for PM10, PM2.5, and PM1 at GCO during 2007 $\sim$ 2012. (Table 1 is uploaded in Fig. 7)

Comment 9: Please add figure to show the species proportion in PM. Response 9: The chemical fraction of PMs is given in Supplementary Information (SI 2), where OC is not converted to OM.

SI 2. The average chemical composition of PM10, PM2.5, and PM1 using all measured species, where OC is not converted to organic matter (OM). Comment 10: Line 153-168, the describe for figure 2 is not accurate Response 10: This comment must be about Figure 1 rather than Figure 2. We recognized the gray shades were in wrong places and corrected them in the revised manuscript, as follows.

Comment 11: PCA analysis of PM, two components are usually selected, but it uses three components in multi-linear regression analysis, why? I don't think it is right to using mass of PM as factor in PCA analysis because mass of PM acts as dependent variable in multi-linear regression analysis. Response 11: In general, two independent factors are extracted from PCA, as you pointed out. The reason for considering the 3rd factor (PC3) is that PC3 exhibits a clear increasing trend with time (Figure shown below). In addition, PC3 loadings were discernible during warm season, in contrast to other two components that were dominated in cold season. Its contribution is 9 %, which is not negligible, either. PC3 loadings of PM2.5 and PM10 were moderately related to NH4+ concentration, indicating agricultural influence. These properties make PC3 to be an individual factor that is orthogonal to the other two factors.

In northeast Asia, the variation of PM mass is intricately intertwined with various types of emissions and physicochemical processes under dynamic meteorological change. That is why we analyzed the three size-cuts of PMs and included all variables for PCA. The following results are good examples. While PM10 mass is highly elevated upon dust outbreak, it was more closely associated with PC1, indicating pollution influence

on PM10 mass (Figure 3). The multiple regression of PC loadings indicates dust factor negatively affected PM1 mass. It can be understood as scavenging effect of dust particles on submicron aerosols.

Comment 12: I think it exists errors if the study didn't exclude dust and haze event in PCA analysis. Response 12: As stated in Response 11, the variation of PM mass of the study region is intricately intertwined with various types of emissions and physico-chemical processes under dynamic meteorological change. If meteorological condition meets, soil particles are transported and mixed with fine aerosols. If its influence is only for a short time period and not visibly discernible, it is not issued as an event. Therefore, it can't be said that there is no influence of soil particles on PMs because it is not a dust day. It won't be a problem if we have measurable criteria for dust or haze. But it is practically difficult. In particular, the EPA's FRM (Federal Reference Method) is defined as a manual sampling of PMs on a filter for 24 hours, which is relatively long, compared to the duration time of weak haze and dust. In this context, we attempted to diagnosis the influence of haze or dust influence using long-term measurements.

Comment 13: Perhaps you maybe use PMF model to analyze source of PM. Response 13: In this study, PM10, PM2.5, and PM1 were simultaneously measured, with which we tried to comprehensively understand the key factors that determined their variations. Principle Component Analysis (PCA) would be suitable for this purpose because it only extracts a few number of orthogonal factors (Chavent et al., 2009). Actually, the PCA is the basis for source apportionment methods such as Positive Matrix Factorization (PMF) or Non-negative Matrix Factorization (NMF). Practically, PMF method requires more variables than those available in this study including metals.

Chavent, M., Guegan, H., Kuentz, V., Patouille, B., and Saracco, J.: PCA‐and PM-F‐based methodology for air pollution sources identification and apportionment, Environmetrics, 20, 928-942, https://doi.org/10.1002/env.963, 2009.

Comment 14: Line 49-52, Please add the instrument condition of analyzing ions. How

about pretreatment of PM samplers for ions analysis and how to quantify the concentrations of ions? Response 14: Do you mean the instrument condition in the previous paper by Shang et al. (2018a)? The condition of the instrument was the same as that of this study. If you mean the condition for chemical analysis, it is given in Methodology section. This section is modified with more details as follows.

Page 5 line 124-130: "Aerosol samples were collected separately for PM1, PM2.5, and PM10 onto 37 mm Teflon and Quartz filters (Pall, Corp.) using sharp-cut cyclones (URG, USA) at the Gosan Climate Observatory (GCO) (33°17′N, 126°10′E) from 2007 to 2012 (Fig. 1). As an ideal location to observe continental outflows in northeast Asia, the GCO has been used as a key measurement site not only for intensive field campaigns such as ABC-EAREX2005 (Atmospheric Brown Cloud–East Asia Regional Experiment) (e.g., Lee, et al., 2007), but also long-term studies (e.g., Lim et al., 2018; Shang et al., 2018a)."

Page 6 line 135-138: "The samples of PM10, PM2.5 and PM1 were concurrently collected approximately every 6–8 days during the five years. As a result, our samples do not cover all officially issued haze and dust events, thereby being suitable for diagnosing the effect of haze and dust on particulate matters."

Page 6 line 143-144: "Water-soluble species were extracted from the filters into a solution comprising a mixture of 19 mL distilled water and 1 mL methanol via 20−min sonication."

Comment 15: what is basis of dust and haze diagnosis based on method using in this study. Response 15: The criteria were given from the statistical analysis of PM10, PM2.5, and PM1 mass measured at GOC for 5 years (Table 3 and Figure 5). The criteria for the impact of dust and haze are the mean+SD, which was corresponding to the upper 10 %. If the daily PM10 and PM1 mass is over mean+SD, it is highly likely to be impacted by dust and haze particles, respectively. It is also highlighted in the present study that PM2.5 is under the influence of dust as well as haze.

It was tested if these criteria were valid for the recent measurements at Gosan from January 2016 to October 2017. As PM2.5 measurement officially began in 2015, the hourly measurements of PM10 and PM2.5 are available through http://www.airkorea.or.kr/. During this period, dust and haze events were observed for eight and twelve days, respectively, for which daily averaged concentrations are presented in Figure 5 (modified). As shown in the Figure, all dust and haze, and haze events are found above the mean+SD of PM10 (52 $\mu$g/m3) and PM2.5 (32 $\mu$g/m3), respectively. It demonstrates that the criteria suggested in this study are robust and useful to diagnose the effect of dust and haze impacted particles.

Figure 5. Frequency distributions of PM10, PM2.5 and PM1 mass concentrations for all measurements. Mass concentrations are given as ln values in x-axis. The green lines stand for mean+SD. The individual samples collected during dust or haze events are marked as different symbols along the x-axis. For comparison, added right below the x-axis are the daily average concentrations of haze and dust days during January 2007 $\sim$ October 2012 (http://www.airkorea.or.kr/).

Comment 16: Conclusions, this section is too long. Response 16: Although we tried hard to get the conclusion shorter, we could not find any part to be dropped out. We would really appreciate it if you point out the part that is not necessary.

Comment 17: Table 1 and 3 can be merged. Response 17: We admit there is overlap between the two tables. After Table 1 is remade according to your comment, it delivers too much information and has no room for inserting the main mode in Table 3. Table 3 summarizes the mode analysis results for the three types of particulate matters, including main mode, median, mean, standard deviation. The comparison of these parameters explicitly demonstrates the episodic occurrence of high PM masses. Thus, Table 3 is left as a separate table in the revised manuscript.

Comment 18: Figure 1, There are no data of winter in 2009 and spring in 2011 and 2012. It is scare of persuasion based on Figure 1 about spring. Response 18: Accord-

ing to your Comment 3, the statistical summary of samples is provided in Supplementary Information (SI 1). Sampling was halted for several reasons such as gusty wind, heavy rain or machine breakdown. There are 3 winter samples in 2009 and 7 spring samples in 2011. In spring, there are 7 to 18 samples each other year.

Please see the file uploaded in Supplement.

Please also note the supplement to this comment:
https://www.atmos-chem-phys-discuss.net/acp-2018-721/acp-2018-721-AC2-supplement.pdf

[Figure]

[Figure]

**Fig. 1.**

[Figure]

**Fig. 2.**

[Figure]

[Figure]

[Figure]

**Fig. 3.**

[Figure]

**Fig. 4.**

[Figure]

**Fig. 5.**

[Figure]

**Fig. 6.**

| | PM$_{10}$ | | | PM$_{2.5}$ | | | PM$_{1}$ | | |
|---|---|---|---|---|---|---|---|---|---|
| | $\bar{x}$ | $\bar{x}+$ SD | 10th–50th–90th | $\bar{x}$ | $\bar{x}+$ SD | 10th–50th–90th | $\bar{x}$ | $\bar{x}+$ SD | 10th–50th–90th |
| Mass | 30 | 52 | 11–24–49 | 19 | 32 | 7–15–35 | 14 | 25 | 4–11–27 |
| Cl$^-$ | 0.8 | 1.7 | 0.09–0.48–2.02 | 0.1 | 0.3 | 0.03–0.08–0.28 | 0.1 | 0.2 | 0.03−0.08−0.16 |
| NO$_3^-$ | 2.1 | 4.1 | 0.55–1.44–4.75 | 1.0 | 2.2 | 0.13–0.55–2.39 | 0.7 | 1.9 | 0.09–0.34–1.64 |
| SO$_4^{2-}$ | 5.5 | 9.5 | 1.32–4.72–11.14 | 4.4 | 7.5 | 1.04–3.49–8.43 | 4.3 | 7.5 | 0.84–3.6–8.05 |
| Na$^+$ | 1.2 | 1.9 | 0.42–0.92–2.14 | 0.4 | 0.7 | 0.13–0.28–0.64 | 0.2 | 0.4 | 0.05–0.11–0.32 |
| NH$_4^+$ | 2.8 | 4.6 | 0.98–2.18–5.09 | 2.4 | 4.0 | 0.83–1.9–4.49 | 2.1 | 3.6 | 0.66–1.72–4.06 |
| K$^+$ | 0.3 | 0.5 | 0.06–0.2–0.5 | 0.2 | 0.4 | 0.03–0.14–0.41 | 0.2 | 0.4 | 0.03–0.11–0.37 |
| Mg$^{2+}$ | 0.2 | 0.3 | 0.05–0.15–0.31 | 0.1 | 0.1 | 0.01–0.05–0.12 | 0.02 | 0.04 | 0.01–0.01–0.04 |
| Ca$^{2+}$ | 0.3 | 0.7 | 0.07–0.16–0.55 | 0.1 | 0.3 | 0.04–0.09–0.2 | 0.1 | 0.2 | 0.02–0.05–0.14 |
| OC | 4.0 | 6.6 | 1.04–2.98–7.48 | 3.4 | 5.7 | 1.22–2.47–6.07 | 2.6 | 4.3 | 0.83–2.09–4.81 |
| OC1 | 0.1 | 0.2 | 0–0.1–0.27 | 0.1 | 0.2 | 0–0.1–0.24 | 0.1 | 0.2 | 0–0.08–0.25 |
| OC2 | 0.8 | 1.3 | 0.28–0.71–1.45 | 0.8 | 1.3 | 0.32–0.65–1.44 | 0.7 | 1.1 | 0.27–0.63–1.22 |
| OC3 | 1.2 | 2.0 | 0.34–0.99–2.19 | 0.9 | 1.5 | 0.32–0.68–1.52 | 0.7 | 1.1 | 0.28–0.59–1.2 |
| OC4 | 0.9 | 1.7 | 0.12–0.64–1.83 | 0.7 | 1.3 | 0.12–0.45–1.34 | 0.4 | 0.8 | 0.09–0.34–1.06 |
| OP | 0.9 | 1.7 | 0.17–0.68–2.15 | 0.9 | 1.6 | 0.15–0.69–1.7 | 0.7 | 1.3 | 0.04–0.51–1.51 |
| EC | 1.5 | 2.9 | 0.38–1.01–3.22 | 1.5 | 2.7 | 0.37–1–2.9 | 1.2 | 2.0 | 0.34–0.98–2.2 |
| EC1 | 1.2 | 2.5 | 0.23–0.72–2.72 | 1.1 | 2.3 | 0.22–0.56–2.45 | 0.8 | 1.5 | 0.12–0.54–1.59 |
| EC2+3 | 0.3 | 0.5 | 0.04–0.28–0.53 | 0.4 | 0.6 | 0.13–0.37–0.59 | 0.4 | 0.6 | 0.16–0.38–0.59 |

* $\bar{x}$ = mean, $\bar{x}+SD$ = mean + standard deviation, 10th 50th 90th percentiles.

**Fig. 7.**